# Integrating long noncoding RNAs and mRNAs expression profiles of response to *Plasmodiophora brassicae* infection in Pakchoi (*Brassica campestris* ssp. *chinensis Makino*)

**Hongfang Zhu**[1,2◉], **Xiaofeng Li**[1,2◉], **Dandan Xi**[1,2], **Wen Zhai**[3], **Zhaohui Zhang**[1,2], **Yuying Zhu**[1,2]*

**1** Horticulture Research Institute, Shanghai Academy of Agricultural Sciences, Shanghai, China, **2** Shanghai Key Lab of Protected Horticultural Technology, Shanghai, China, **3** East China University of Technology, Nanchang, China

◉ These authors contributed equally to this work.
* yy5@saas.sh.cn

**Data Availability Statement:** All relevant data is within the National Center for Biotechnology Information Sequence Read Archive under the

## Abstract

The biotrophic protist *Plasmodiophora brassicae* causes serious damage to Brassicaceae crops grown worldwide. However, the molecular mechanism of the *Brassica rapa* response remains has not been determined. Long noncoding RNA and mRNA expression profiles in response to *Plasmodiophora brassicae* infection were investigated using RNA-seq on the Chinese cabbage inbred line C22 infected with *P. brassicae*. Approximately 5,193 mRNAs were significantly differentially expressed, among which 1,345 were upregulated and 3,848 were downregulated. The GO enrichment analysis shows that most of these mRNAs are related to the defense response. Meanwhile, 114 significantly differentially expressed lncRNAs were identified, including 31 upregulated and 83 downregulated. Furthermore, a total of 2,344 interaction relationships were detected between 1,725 mRNAs and 103 lncRNAs with a correlation coefficient greater than 0.8. We also found 15 *P. brassicae*related mRNAs and 16 lncRNA interactions within the correlation network. The functional annotation showed that 15 mRNAs belong to defense response proteins (66.67%), protein phosphorylation (13.33%), root hair cell differentiation (13.33%) and regulation of salicylic acid biosynthetic process (6.67%). KEGG annotation showed that the vast majority of these genes are involved in the biosynthesis of secondary metabolism pathways and plant-pathogen interactions. These results provide a new perspective on lncRNA-mRNA network function and help to elucidate the molecular mechanism of *P. brassicae* infection.

## Introduction

Clubroot, a soil-borne disease, has caused considerable damage to Brassicaceae crops [1, 2]. This disease is caused by the protist *Plasmodiophora brassicae* (P. brassicae), which can survive for up to 20 years in soil [3]. The two stages of P. brassicae, root-hair infection and cortical

accession number PRJNA528807. The Brassica rapa reference genome and gene model annotation files were downloaded from the Genome Database (http://brassicadb.org/).

**Funding:** This work was supported by Shanghai Agriculture Applied Technology Development Program, China (Grant No. G2014070108), Agriculture Research System of Shanghai, China (Grant No. 201903) and National Key R&D Program of China 2017YFD0101803. The funders had no role in study design, data collection and analysis, decision to publish, or preparation of the manuscript.

**Competing interests:** The authors have declared that no competing interests exist.

infection, play an important role in the infection process and make it difficult to control [4]. The Pakchoi (*Brassica campestris ssp. chinensis Makino*), also called non-heading Chinese cabbage, is one of the most important *Brassica* vegetable crops in China and other eastern Asian countries. Most Pakchoi cultivars are highly susceptible to the *P. brassicae*.

To date, considerable progress has been made in cultivating clubroot resistant (CR) crops. Genetic analysis and QTL mapping have identified some CR genes or loci in *Brassica* crops: *CRa* [5], *Crr1a* and *Crr1b* [6], *CRb* [7], *Crr2* [8], *Crr3* [9, 10], *Crr4* [11], *CRc* and *CRk* [12], *Rcr1* [13, 14], *PbBa3.1* and *PbBa3.3* [15], *QS_B1.1* [16], and *Pb-Br8* [17]. Three loci for clubroot resistance, *Rcr4, Rcr8, Rcr9*, have been revealed by Genotyping-by-sequencing, but they cannot be distinguished from the abovementioned loci [18]. Among them, *CRa, Crr1a* and *CRb* have been cloned. *CRa* and *Crr1a* contain Toll-interleukin receptor (TIR)—nucleotide-binding (NB)—leucine -rich repeats (LRRs) and *CRb* contains NB-LRRs, which are known to be responsible for race-specific resistance in higher plants [19, 20]. However, these genes or loci have been demonstrated to be responsible for race-dependent resistance [21], and the molecular mechanism of the *Brassica rapa* responsehas not been determined.

To date, a number of transcriptome sequencing projects have been employed to explore the molecular basis of the interaction between *Brassica* crops and *P. brassicae*. Twenty protein spots that were observed with changes in expression played a role in lignin synthesis, cytokinin synthesis, calcium steady-state, glycolysis, and oxygen activity in *Brassica napus* [22]. Then, the signaling and metabolic activity of jasmonate acid (JA) and ethylene (ET) were found to be upregulated significantly in resistant populations while genes involved in salicylic acid metabolic (SA) and signaling pathways were generally not elevated at 15 days post inoculation (dpi) [13]. Moreover, genes associated with pathogen-associated molecular patterns (PAMPs) and effector recognition, calcium ion influx, hormone signaling, pathogenesis-related (PR) genes, transcription factors, and cell wall modification showed different expression patterns between CR and clubroot-susceptible (CS) lines in *Brassica rapa* [23]. PR genes are involved in SA signaling which is important to clubroot resistance at the early stage after inoculation. In addition, it was proven that response changes in transcript levels under *P. brassicae* infection were primarily activated at the primary stage between Broccoli (*Brassica oleracea* var. *italica*) and wild Cabbage (*Brassica macrocarpa Guss.*) [24]. By comparing the transcriptome landscape between CS and CR Chinese cabbage lines, Jia et al. (2017) confirmed that the differentially expressed genes related to disease-resistance in CR lines enriched in calcium ion influx, glucosinolate biosynthesis, cell wall thickening, SA homeostasis, chitin metabolism and PR pathway. The upregulated genes in CS lines were mostly related to cell cycle control, cell division and energy production and conversion [2]. In addition, the Indole acetic acid (IAA) and cytokinin-related genes were found to affect the root swelling in clubroot development [2, 25].

LncRNAs are a set of RNA transcripts (>200 nt in length) which have no protein-coding ability. During the past several decades, a small number of long noncoding RNAs (lncRNAs) have been identified and shown to mediate various biological processes in plants [26], such as biotic and abiotic stress responses [27, 28]. In plant-pathogen interactions, some lncRNAs have been identified and shown to respond to (1) stripe rust pathogen stress in wheat [24]; (2) Fusarium oxysporum infection [29] and Pseudomonas syringe pv tomato DC3000 (ELF18-induced lncRNA) [30] in *Arabidopsis thaliana*; (3) *Pectobacterium carotovorum* in potato [31]; (4) *Phytophthora infestans* (lncRNA 16397) [32], tomato yellow leaf curl virus [33] and *Phytophthora infestans* in tomato (lncRNA23468) [34]; and (5) *Sclerotinia sclerotiorum* in *Brassica napus* [35]. The *B. rapa* and *B. napus* genome has a large number of lncRNAs [36, 37]. In addition, lncRNAs are demonstrated with the ability to be expressed broadly across many developmental times and in different tissue types [37]. However, only a few lncRNAs coexpressed with genes of temperature expression patterns were reported in *Brassica rapa* [36].

In this study, we first conducted a comprehensive analysis of intergrating long noncoding RNAs and mRNA expression profiles of response to *Plasmodiophora brassicae* infection in *Brassica rapa* L. and identified a great number of significant differentially expressed genes and some lncRNAs. The regulatory network of mRNA and lncRNA helps to elucide the *Brassica rapa* responses during *P. brassicae* infection and breeding of resistant CR cultivars.

## Materials and methods

### Ethics statement

This study was carried out in a phytotron. No specific permissions were required. The study did not involve any endangered or protected species.

### Sample collection

Pakchoi inbred line CS22 is a cold tolerant type and susceptible to the 7th physiology race of *Plasmodiophora Brassicae* by using the inoculation method of Williams [38]. The pathogen was propagated on CS22 named CS22A, and the clubs in infected roots were stored at -20˚C until required. All plants were sown in a growth chamber at 25/20˚C (day/night) with a photo-period of 14h containing. The CS22A plants were inoculated in a pot containing $5 \times 10^6$ spores per gram of dry soil. The root tissue samples were obtained by 6 weeks post inoculation. No infected root samples of CS22 were the control. For each treatment, the samples were immediately frozen in liquid nitrogen and then stored at −80˚C until use. All plant materials examined in this study were obtained from Shanghai Academy of Agricultural Sciences.

### RNA extraction, library construction, and sequencing

Total RNA was extracted from each root tissue sample using the mirVana miRNA Isolation Kit (Ambion) following the manufacturer's protocol. RNA integrity was evaluated using the Agilent 2100 Bioanalyzer (Agilent Technologies, Santa Clara, CA, USA). The samples with RNA Integrity Number (RIN) $\geq$ 7 were subjected to the subsequent analysis. The libraries were constructed using TruSeq Stranded Total RNA with Ribo-Zero Gold according to the manufacturer's instructions. The main steps of library construction and sequencing are as follows: (1) removing rRNA from total RNA, (2) breaking RNA into fragments, (3) RNA fragments are reverse-transcribed into cDNA, (4) adapter sequences are added to cDNA, and suitable fragment sizes are selected for the next step, and (5) PCR amplification. Then these libraries were sequenced in the Illumina HiSeq[TM] 2500 sequencing platform and 150 bp paired-end reads were generated.

### Data filtering and transcriptome assembly

The RNA-seq data sets were analyzed as previously described [39]. High quality clean data were kept for downstream analysis after we use Trimmomatic v0.32 with 'LEADING:3 TRAILING:3 SLIDINGWINDOW:4:15 MINLEN:50' to remove low-quality reads from the raw data, such as the reads containing adapters, the reads containing over 10% of poly (N), and low-quality reads ($>$ 50% of the bases having Phred quality scores $<$10). Basic information of clean data was calculated, such as read number, base contents, Phred score (Q30) and GC content. *Brassica rapa* reference genome and gene model annotation files, which were downloaded from the Genome Database (http://brassicadb.org/). First, the index of the reference genome was built with Hisat-build, and then paired-end clean reads were aligned to the reference genome using Hisat with default parameters [40]. Second, the sam format result from hisat2 was translated to bam format by samtools [41], and then bam files from each library were

assembled with Stringtie [42]. Stringtie was run with '-library-type fr-firststrand', and other parameters were set as default. Last, each library results from Stringtie were merged to a final genome transcript feature file by cuffmerge [43].

## Pipeline for LncRNA identify

To obtain putative lncRNAs, assembled novel transcripts were filtered following the steps according to the assembly results. (1) First, cuffcompare was used to compare the assembly transcript and reference transcript one by one [43], only transcripts annotated as "i", "u", "x", and "o" representing a transfrag falling entirely within a reference intron, unknown intergenic transcript, exonic overlap with reference on the opposite strand, and generic exonic overlap with a reference transcript, respectively, were retained. (2) Second, the transcripts with a length of above 200 bp and with an exon number of more than 1 were kept for the next step. (3) Finally, four different methods were used to identify the coding potential of new transcripts, namely, Coding Potential Calculator (CPC) [44], Coding-Non-Coding Index (CNCI) [45], PLEK [46] and Pfam [47]. The methods were used to assess the coding potential of the remaining transcripts from step 2. Transcripts that were likely to contain a known protein-coding domain removed. Only transcripts considered to be lncRNAs via four methods will be kept for downstream analysis.

## Identification of differentially expressed mRNA and lncRNA

Express [48] and bowtie2 [49] were used to calculate FPKM scores for the lncRNAs and coding genes in each library. Differentially expressed lncRNAs and mRNAs between any two libraries were identified by DESeq (release 3.2) [50]. P value < 0.05 and an absolute value of the fold change $\geq 2$ were used as a threshold to evaluate the statistical significance of lncRNA and mRNA expression differences.

## Quantitative real-time PCR validation

To validate the credibility of the findings of RNA analysis, mRNAs and lncRNAs were randomly selected for real-time PCR. Total RNA was collected from the root tissue samples of the two groups using TruSeq Stranded Total RNA LT—(with Ribo-Zero Plant). The SuperScript III First-Strand Synthesis System was used to reverse the transcription to cDNA. Quantitative RT-PCR was conducted in a ViiA 7 Real-time PCR System (Applied Biosystems) using PowerUp™ SYBR Green Master Mix (Applied Biosystems, Carlsbad, CA, USA). Tubulin beta-6 (TUB6) was used as an internal control to normalize the data [51]. The primers used in qRT-PCR and cDNA synthesis were designed in the laboratory and synthesized by OEBiotech (Shanghai OEBiotech. Co., Ltd, Shanghai, China) based on the sequences. Primers are listed in S1 Table. The reaction conditions were as follows: incubation at 95˚C for 10 min, followed by 40 cycles of 95˚C for 10 s and 60˚C for 1 min. The relative expression levels were calculated using the $2^{-\Delta\Delta Ct}$ method and were normalized to *TUB6*, as an endogenous reference transcript.

## Functional enrichment of differentially expressed mRNA

The Gene Ontology (GO) database (http://www.geneontology.org) is a description database that was usually applied to elucidate the genetic regulatory network of interest by forming hierarchical categories according to the molecular function, biological process, and cellular component. The Kyoto Encyclopedia of Genes and Genomes (KEGG, http://www.genome.jp/kegg/) is the main public database about pathways. GO annotation and pathway analysis were

used to study the effects of all significant differentially expressed mRNAs. The p value is calculated using hypergeometric test. R program packages were used to elucidate the GO and KEGG for targets of significant differential enrichment. GO and KEGG terms with P values < 0.05 were recognized as significant enrichment.

### Target gene prediction

The function of lncRNAs is mainly realized by cis acting on target genes. The basic principle of cis-acting target gene prediction holds that the function of lncRNA is related to the protein-coding genes adjacent to its coordinates; therefore, the mRNA adjacent to lncRNA is selected as its target gene. Target gene analysis method: Pearson correlation coefficients of lncRNA and mRNA ≥ 0.8 were required. LncRNA is determined as regulator if it is within 100 k upstream or within 100 k downstream of mRNAs.

### LncRNA-mRNA co-expression network construction

According to the differentially expressed lncRNA and mRNA results, we constructed a regulatory network to identify the relationships between lncRNA genes and mRNA genes. The Pearson correlation test was used to calculate the correlation between differential lncRNA and mRNA expression data. Pearson's correlation coefficients equal to or greater than 0.8 and a P value less than 0.05 were considered to be lncRNA-mRNA pairs. Arranging from small to large according to the p-value, we chose 600 top results to construct the regulatory network, and the lncRNA-mRNA pairs associated with disease resistance were also used for the network construction. Cytoscape software (Cytoscape Consortium, San Diego, CA, USA) was used to present the lncRNA-mRNA regulatory network relationship.

### Accession number

The RNA-seq datasets used in this study can be found in the NCBI Gene Expression Omnibus under accession number: PRJNA528807.

## Results

### Overview of RNA sequencing

To elucidate lncRNA and mRNA expression patterns in response to *Plasmodiophora brassicae* infection in *Brassica rapa* L., 6 libraries were constructed from control and clubroot tissues (CS22A) (Fig 1) for three biological replicates and sequenced using the Illumina platform [52]. The raw data obtained from RNA-seq are available in the National Center for Biotechnology Information (NCBI) Sequence Read Archive (SRA). A total of 296.14 million and 295.29 million raw data reads were obtained in the control and clubroot libraries, respectively. The number of reads after quality filtering of the above two libraries were 289.57 million and 289.73 million respectively. Approximately 61.45% of the reads were mapped to the reference genome (BRAD database http://brassicadb.org/) [53]. Q30 (reads with an average quality score > 30) reads were more than 93% and the GC content of all sequencing libraries were less than 52%. In this study, we identified 38,483 mRNAs and 1,492 lncRNAs in the control and clubroot libraries. Gene expression levels were calculated using the FPKM (fragments per kilobase of exon model per million mapped) method [54]. Among the identified lncRNAs, 659 lncRNAs were known based on the database of CANTATAdb 2.0 (http://cantata.amu.edu.pl/) [55]. The mRNA expression level varied from 0 to 9,527.5 among all libraries, with an average value of 20.2. The lncRNA expression level varied from 0 to 65,092.9 among the six libraries, with an average value of 218.5. Principle component analysis (PCA) of the transcriptome and lncRNA

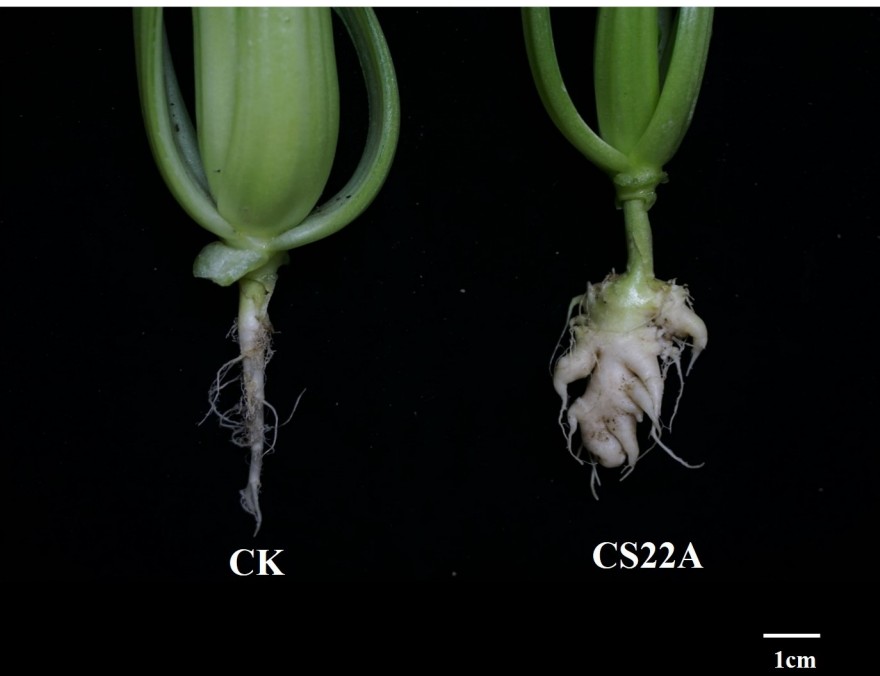

**Fig 1. Clubroot symptoms of CS22.** Plants were inoculated with the 7th physiology race of *P. brassicae* (CS22A), while the control was not subjected to pathogen inoculation.

FPKM values for all samples showed more separation between control and clubroot samples, respectively. This result showed that the sequencing data could be used for further analysis.

## mRNA and lncRNA expression profiles in Pakchoi

Compared to the control samples, 5,193 mRNAs were observed to be significantly differentially expressed (fold change $\geq$ 2 and P $\leq$ 0.05), including 1,345 upregulated and 3,848 downregulated in CS22A. In total, a number of 114 significantly differentially expressed lncRNAs were identified, including 31 upregulated and 83 downregulated. The number of downregulated mRNAs and lncRNAs was higher than the number of upregulated. Clustering analysis of the top 40 most significantly differentially expressed mRNAs between control and CS22A is shown with a heatmap (Fig 2A, S2 Table), and the heatmap of the top 40 significantly differentially expressed lncRNAs is shown in Fig 2B

The length distribution and categorization of identified lncRNAs were also analyzed (Fig 3). The length of lncRNAs ranged from 200 to 4,483 bp, with an average length of 658 bp. The most abundant lncRNAs were between 200–400 bp. The number of lncRNAs decreased as the length increased. The lncRNA lengths were mostly less than 2,000 bp. lncRNAs were categorized into four groups, intronic, intergenic, sense and antisense based on their location on the genome [56, 57]. The majority of lncRNAs (55.16%) were intergenic and located in intergenic regions. The rates of lncRNAs were 2.41%, 27.82% and 14.61% for intronic, sense and antisense, respectively. Because lncRNAs encode small RNAs, the sequences of the lncRNAs were mapped to small RNA precursors. Twenty-five small RNA families were mapped to fifteen lncRNAs.

To confirm the expression level of differentially expressed RNAs identified from the RNA sequencing data, qRT-PCR analysis was used to assay the expression level of 10 randomly selected differentially expressed RNAs and lncRNAs. The trend of expression changes of these

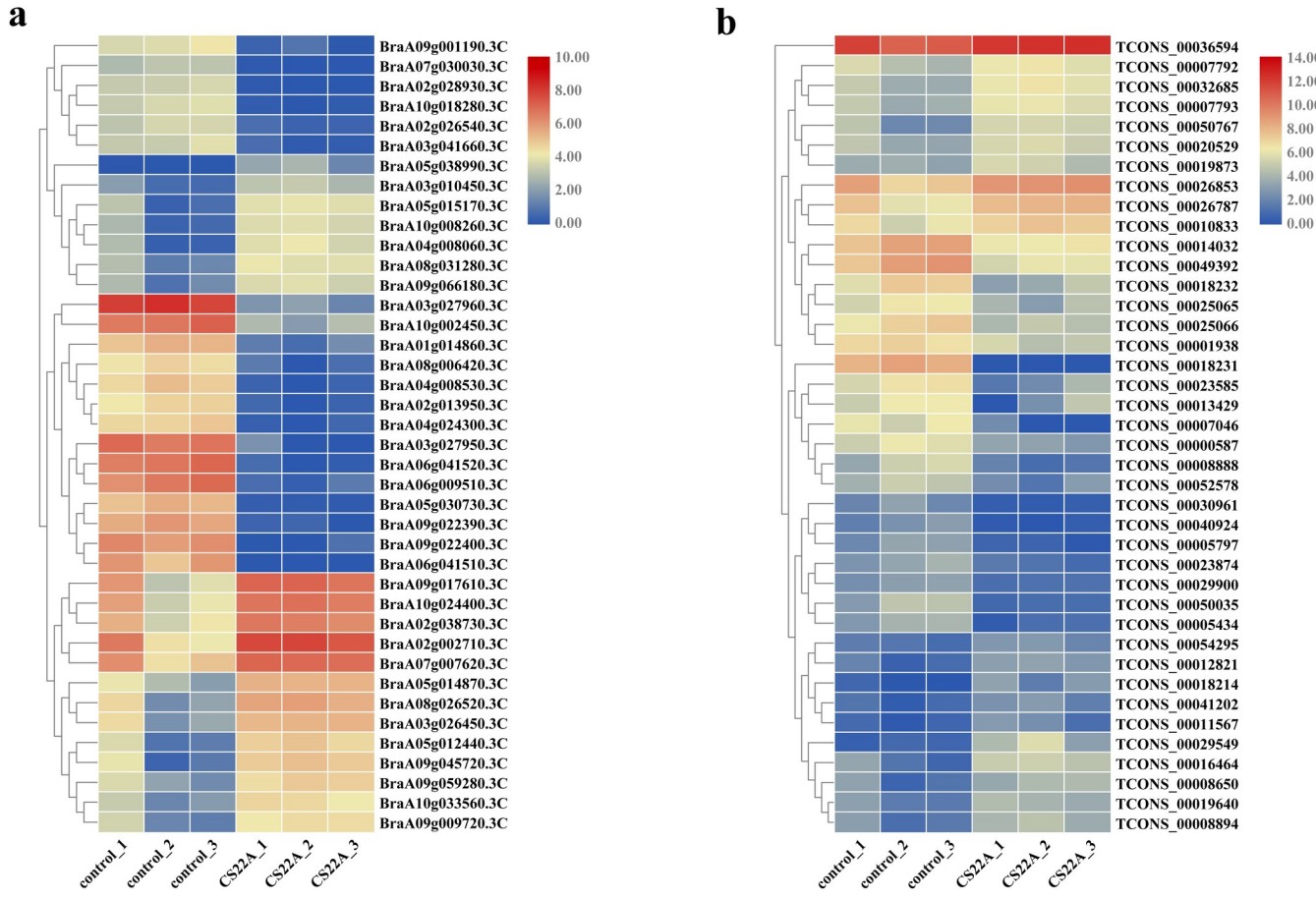

**Fig 2.** Heatmap of 40 significantly differentially expressed mRNAs (A) and lncRNAs (B).

select genes based on the qRT-PCR was similar to the sequencing data, which suggested that the RNA-seq data were reliable (Fig 4).

## Functional annotation analysis of significant differentially expressed mRNAs

GO enrichment analysis was conducted on the significantly differentially expressed mRNAs (fold change ≥ 2 and P ≤ 0.05) to gain more insights into the function of these mRNAs which can be divided into three main functional groups (Fig 5). In biological processes, the top 40 GO terms of the upregulated mRNAs showed that the majority of the functions related to the defense response to bacterium (GO:0042742), the defense response to fungus (GO:0050832), the response to wounding (GO:0009611), the response to jasmonic acid (GO:0009753) and the response to toxic substances (GO:0009636) (Fig 5, S3 Table). GO categories of the downregulated genes were shown to be closely related to defense response (GO:0006952), defense response to bacterium (GO:0042742), auxin-activated signaling pathway (GO:0009734), response to wounding (GO:0009611), response to auxin (GO:0009733) and response to jasmonic acid (GO:0009753) (Fig 5, S3 Table). It can be assumed that the genes or proteins that the mRNAs code for are involved in the reaction. In the cellular component, upregulated genes were mapped to membrane (GO:0016020), thylakoid (GO:0009579 and GO:0044436) and membrane protein complex (GO:0098796), while downregulated genes were mapped to

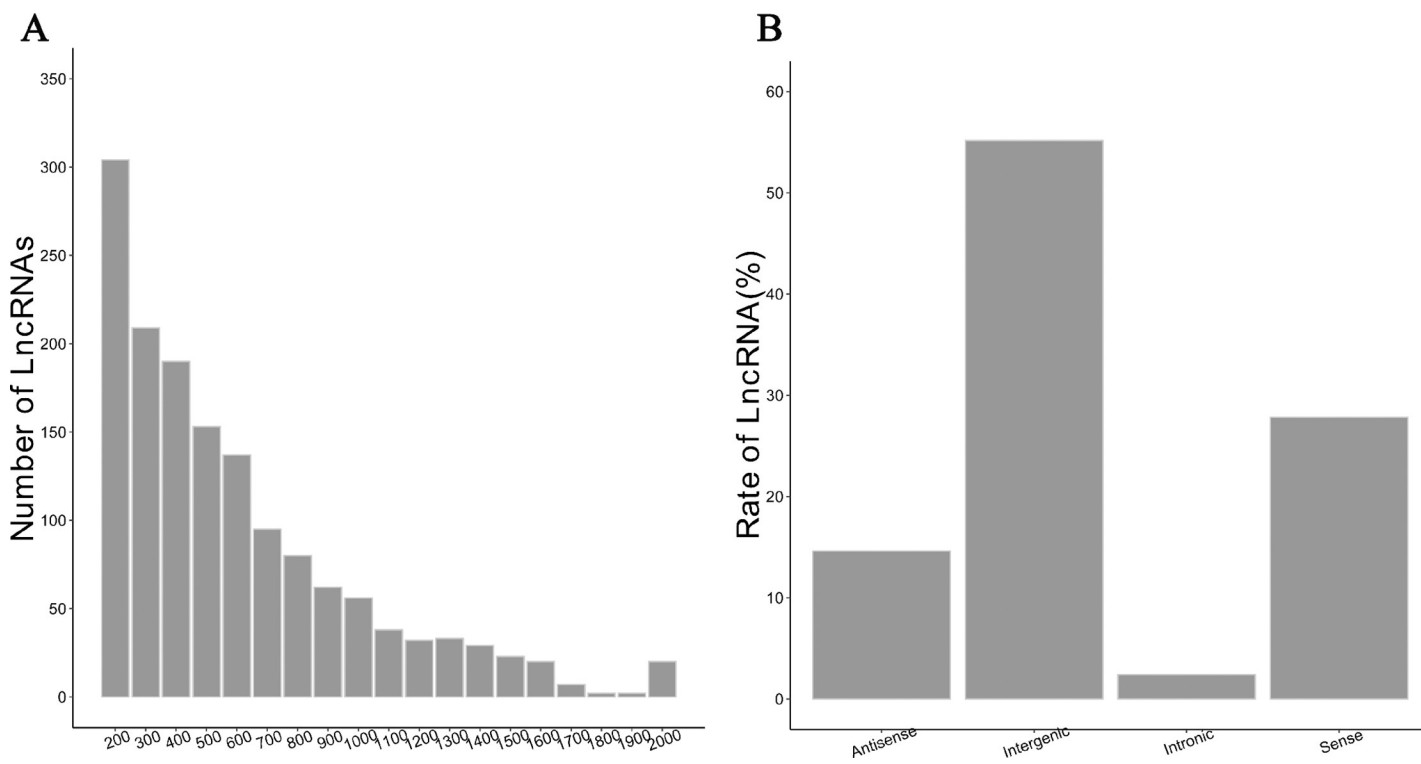

**Fig 3. Length distribution and categorization of identified lncRNAs.** (A) The length distribution of identified lncRNAs. X-axis: the length of LncRNAs. (B) The rate of lncRNAs based on their location on the genome.

intrinsic component of membrane (GO:0031224), integral component of membrane (GO:0016021), and membrane (GO:0044425 and GO:0016020). Regarding the molecular function, the enriched GO terms targeted by upregulated genes included catalytic activity (GO:0003824), oxidoreductase activity (GO:0016491), cofactor binding (GO:0048037) and transporter activity (GO:0005215), the enriched GO terms targeted by downregulated genes included catalytic activity (GO:0003824), transferase activity (GO:0016740) and oxidoreductase activity (GO:0016491).

Kyoto Encyclopedia of Genes and Genomes (KEGG) pathway enrichment analysis of the significantly differentially expressed mRNAs indicated that the top 3 KEGG terms for downregulated mRNAs were associated with plant hormone signal transduction (ko04075), MAPK (mitogen-activated protein kinase) signaling pathway (ko04016) and ABC (ATP-binding cassette transporters) transporters (ko02010), while the top 3 KEGG terms for upregulated mRNAs were associated with biofilm formation (ko02026), drug metabolism (ko00982) and metabolism of xenobiotics by cytochrome (ko00980). The plant hormone signal transduction pathway included 8 plant hormones that contained such acides as jasmonic acid and salicylic acid (https://www.genome.jp/dbget-bin/www_bget?ko04075). Therefore, these genes likely play an important role in the interaction in the infected process.

### Target analysis for *cis*-regulated lncRNAs and their function annotation in significantly differentially expressed lncRNAs

Previous studies reported that lncRNAs regulated neighboring or overlapping genes and might show linked function or co-expression with their target genes [58–60]. Significant differentially expressed mRNAs located within 100 kb windows upstream or downstream of the

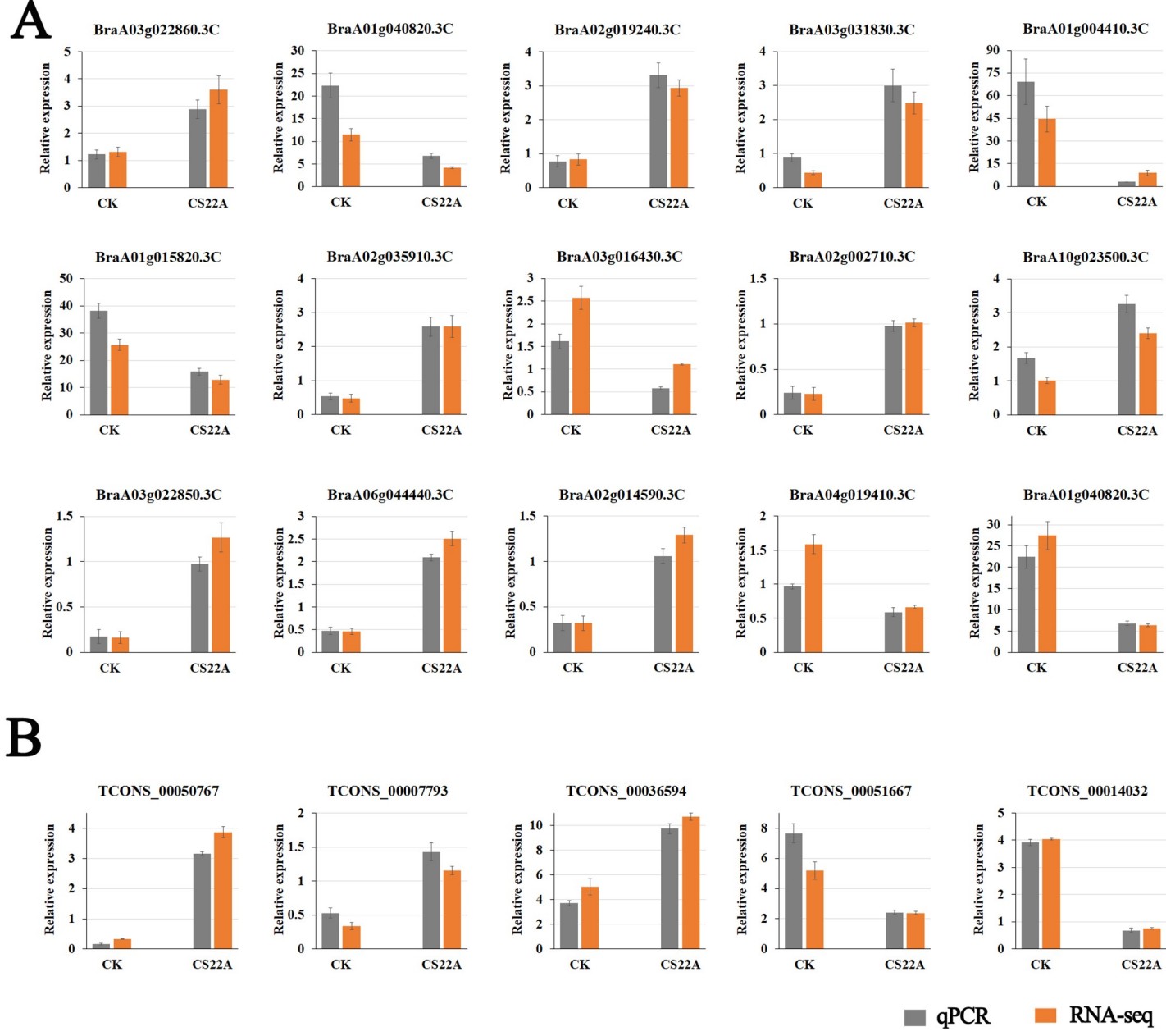

**Fig 4. Randomly selected differentially expressed RNAs were analyzed using qRT-PCR. The expression level was normalized using *TUB6*.** Y-axis: the relative expression of selected genes compared with control as indicated. Data are shown as the mean ± standard deviation of three independent experiments.

lncRNAs were used to calculate the Pearson correlation coefficient for further analysis. A total of 2,344 interaction relationships (1,479 positive and 865 negative correlation) were detected between 1,725 mRNA and 103 lncRNA with a correlation coefficient greater than 0.8. The GO analysis was based on biological processes for all potential target mRNAs. Functional analysis showed that the upregulated co-expressed mRNAs of the neighboring lncRNAs were enriched in 39 GO terms in biological processes, and many of the GO terms were closely related to the regulation of gene expression (Table 1). The downregulated mRNAs of the potential lncRNA targets showed that the term GO enrichment was closely related to the response to stimulus (GO:0050896), response to stress (GO:0006950), defense response (GO:0006952) and response

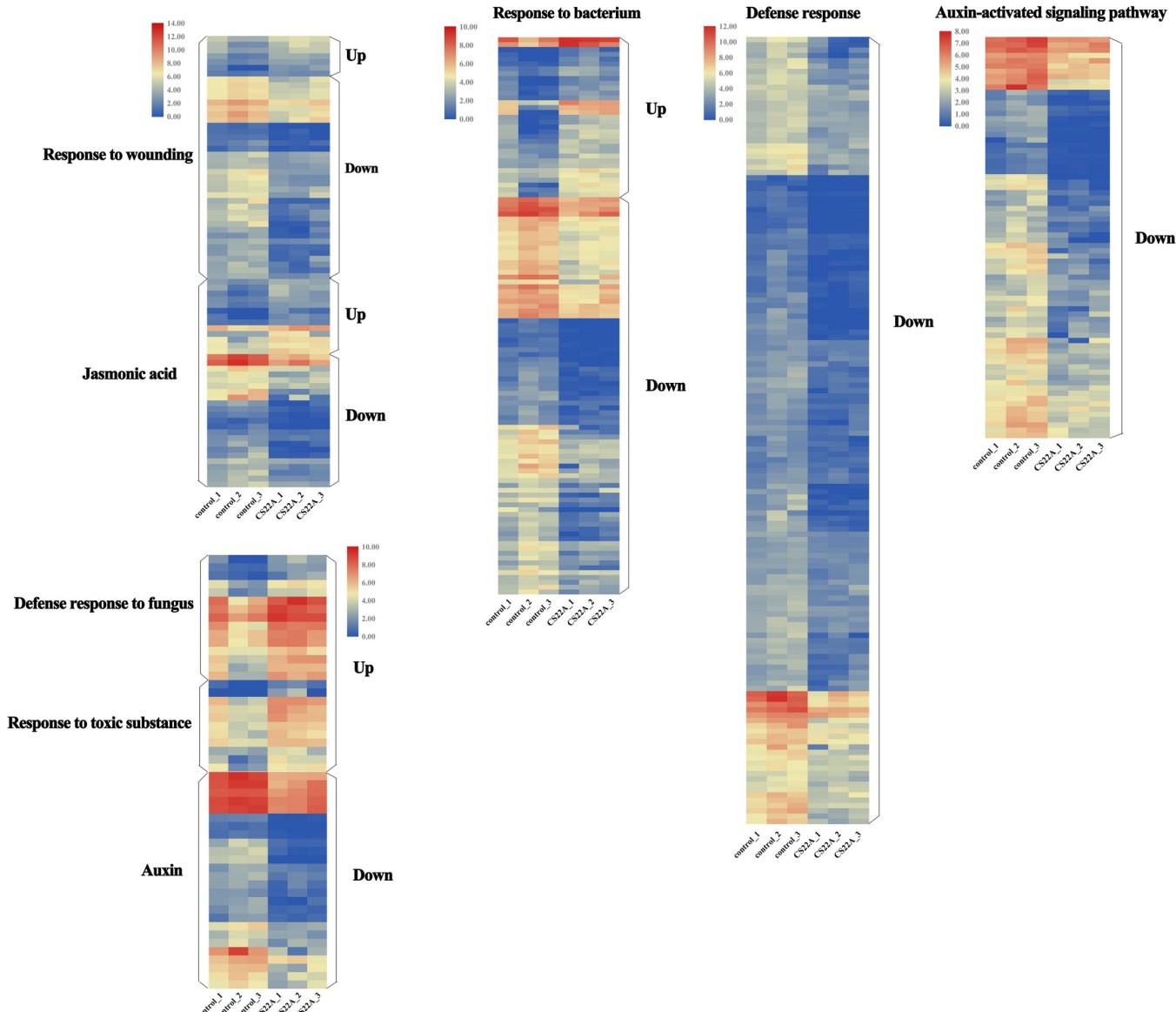

**Fig 5. Heatmaps of significantly differentially expressed mRNAs classified by biological process.**

to biotic stimulus (GO:0009607) (Table 2). Regarding the molecular function, the enriched GO terms targeted by upregulated genes included oxidoreductase activity (GO:0016491), cofactor binding (GO:0048037), and transmembrane transporter activity (GO:0022857), and the enriched GO terms targeted by downregulated genes included catalytic activity (GO:0003824), transferase activity (GO:0016740), and oxidoreductase activity (GO:0016491). In the cellular component, the upregulated gene showed that the majority of the function related to membrane protein complex (GO:0098796) and thylakoid (GO:0044436 and GO:0009579). GO categories of the downregulated genes were shown to be closely related to integral component of membrane (GO:0016021), intrinsic component of membrane (GO:0031224), and membrane (GO:0044425 and GO:0016020). The expression levels of the downregulated mRNAs for the above four GO terms (40 mRNAs) and 16 lncRNAs that regulated these RNAs are shown in Fig 6 and S4 Table, and all these mRNAs and lncRNAs were

**Table 1. Gene Ontology (GO) enrichment for the significantly upregulated co-expressed mRNAs of the neighboring lncRNAs.**

| GO ID | Term | Annotated | Significant | Expected | Classic Fisher |
|---|---|---|---|---|---|
| GO:0015979 | photosynthesis | 122 | 9 | 1.65 | 3.90E-05 |
| GO:0006091 | generation of precursor metabolites and energy | 75 | 7 | 1.01 | 6.50E-05 |
| GO:0009765 | photosynthesis, light harvesting | 34 | 5 | 0.46 | 8.50E-05 |
| GO:0019684 | photosynthesis, light reaction | 51 | 5 | 0.69 | 0.0006 |
| GO:0009414 | response to water deprivation | 9 | 2 | 0.12 | 0.0061 |
| GO:0055114 | oxidation-reduction process | 1431 | 30 | 19.31 | 0.0088 |
| GO:0019438 | aromatic compound biosynthetic process | 1660 | 33 | 22.4 | 0.0129 |
| GO:0018130 | heterocycle biosynthetic process | 1667 | 33 | 22.49 | 0.0136 |
| GO:0006811 | ion transport | 485 | 13 | 6.54 | 0.0141 |
| GO:0005985 | sucrose metabolic process | 14 | 2 | 0.19 | 0.0148 |
| GO:1901362 | organic cyclic compound biosynthetic process | 1677 | 33 | 22.63 | 0.0148 |
| GO:0006355 | regulation of transcription, DNA-template | 1507 | 30 | 20.33 | 0.0175 |
| GO:1903506 | regulation of nucleic acid-templated transcription | 1507 | 30 | 20.33 | 0.0175 |
| GO:2001141 | regulation of RNA biosynthetic process | 1507 | 30 | 20.33 | 0.0175 |
| GO:0051252 | regulation of RNA metabolic process | 1512 | 30 | 20.4 | 0.0183 |
| GO:0010556 | regulation of macromolecule biosynthetic | 1513 | 30 | 20.42 | 0.0184 |
| GO:2000112 | regulation of cellular macromolecule biosynthetic | 1513 | 30 | 20.42 | 0.0184 |
| GO:0009889 | regulation of biosynthetic process | 1515 | 30 | 20.44 | 0.0187 |
| GO:0031326 | regulation of cellular biosynthetic process | 1515 | 30 | 20.44 | 0.0187 |
| GO:0019219 | regulation of nucleobase-containing compound metabolic process | 1517 | 30 | 20.47 | 0.019 |
| GO:0034654 | nucleobase-containing compound biosynthetic | 1649 | 32 | 22.25 | 0.0198 |
| GO:0051171 | regulation of nitrogen compound metabolic | 1538 | 30 | 20.75 | 0.0226 |
| GO:0080090 | regulation of primary metabolic process | 1540 | 30 | 20.78 | 0.023 |
| GO:0009415 | response to water | 18 | 2 | 0.24 | 0.024 |
| GO:0031323 | regulation of cellular metabolic process | 1549 | 30 | 20.9 | 0.0247 |
| GO:0006487 | protein N-linked glycosylation | 2 | 1 | 0.03 | 0.0268 |
| GO:0009072 | aromatic amino acid family metabolic process | 2 | 1 | 0.03 | 0.0268 |
| GO:0006351 | transcription, DNA-templated | 1630 | 31 | 21.99 | 0.0281 |
| GO:0032774 | RNA biosynthetic process | 1630 | 31 | 21.99 | 0.0281 |
| GO:0097659 | nucleic acid-templated transcription | 1630 | 31 | 21.99 | 0.0281 |
| GO:0010468 | regulation of gene expression | 1599 | 30 | 21.58 | 0.0361 |
| GO:0006772 | thiamine metabolic process | 3 | 1 | 0.04 | 0.0399 |
| GO:0009228 | thiamine biosynthetic process | 3 | 1 | 0.04 | 0.0399 |
| GO:0042723 | thiamine-containing compound metabolic process | 3 | 1 | 0.04 | 0.0399 |
| GO:0042724 | thiamine-containing compound biosynthetic | 3 | 1 | 0.04 | 0.0399 |
| GO:0044070 | regulation of anion transport | 3 | 1 | 0.04 | 0.0399 |
| GO:0060255 | regulation of macromolecule metabolic process | 1624 | 30 | 21.91 | 0.0431 |
| GO:0019222 | regulation of metabolic process | 1635 | 30 | 22.06 | 0.0466 |
| GO:0006812 | cation transport | 350 | 9 | 4.72 | 0.0475 |

Annotated: number of genes that are annotated with the GO-term.

Significant: number of genes belonging to the term that are annotated with the GO-term.

Expected: an estimate of the number of genes a node of size annotated would have if the significant genes were to be randomly selected from the gene universe.

Classic Fisher: p-values computed by Fisher's exact test

significantly differentially expressed between the control and clubroot groups. These results suggest that the principal functions of these lncRNAs may be the regulation of gene expression and play an important role in the clubroot infection process.

**Table 2. Gene Ontology enrichment for the significantly downregulated co-expressed mRNAs of the neighboring lncRNAs.**

| GO ID | Term | Annotated | Significant | Expected | classicFisher |
|---|---|---|---|---|---|
| GO:0006979 | response to oxidative stress | 136 | 20 | 5.22 | 2.20E-07 |
| GO:0006950 | response to stress | 552 | 42 | 21.18 | 1.60E-05 |
| GO:0032989 | cellular component morphogenesis | 10 | 4 | 0.38 | 0.00037 |
| GO:0055114 | oxidation-reduction process | 1431 | 78 | 54.9 | 0.00073 |
| GO:0050896 | response to stimulus | 1190 | 67 | 45.65 | 0.00078 |
| GO:0048869 | cellular developmental process | 14 | 4 | 0.54 | 0.00157 |
| GO:0001558 | regulation of cell growth | 8 | 3 | 0.31 | 0.00272 |
| GO:0009826 | unidimensional cell growth | 8 | 3 | 0.31 | 0.00272 |
| GO:0042814 | monopolar cell growth | 8 | 3 | 0.31 | 0.00272 |
| GO:0051510 | regulation of unidimensional cell growth | 8 | 3 | 0.31 | 0.00272 |
| GO:0051513 | regulation of monopolar cell growth | 8 | 3 | 0.31 | 0.00272 |
| GO:0060560 | developmental growth involved in morphogenesis | 8 | 3 | 0.31 | 0.00272 |
| GO:0000902 | cell morphogenesis | 9 | 3 | 0.35 | 0.00396 |
| GO:0022603 | regulation of anatomical structure morphogenesis | 9 | 3 | 0.35 | 0.00396 |
| GO:0022604 | regulation of cell morphogenesis | 9 | 3 | 0.35 | 0.00396 |
| GO:0009653 | anatomical structure morphogenesis | 19 | 4 | 0.73 | 0.00523 |
| GO:0031667 | response to nutrient levels | 10 | 3 | 0.38 | 0.0055 |
| GO:0031669 | cellular response to nutrient levels | 10 | 3 | 0.38 | 0.0055 |
| GO:0040008 | regulation of growth | 10 | 3 | 0.38 | 0.0055 |
| GO:0048589 | developmental growth | 10 | 3 | 0.38 | 0.0055 |
| GO:0048638 | regulation of developmental growth | 10 | 3 | 0.38 | 0.0055 |
| GO:0006952 | defense response | 155 | 13 | 5.95 | 0.00664 |
| GO:0008272 | sulfate transport | 22 | 4 | 0.84 | 0.00902 |
| GO:0072348 | sulfur compound transport | 22 | 4 | 0.84 | 0.00902 |
| GO:0015698 | inorganic anion transport | 63 | 7 | 2.42 | 0.01011 |
| GO:0009991 | response to extracellular stimulus | 13 | 3 | 0.5 | 0.01203 |
| GO:0031668 | cellular response to extracellular stimulus | 13 | 3 | 0.5 | 0.01203 |
| GO:0071496 | cellular response to external stimulus | 13 | 3 | 0.5 | 0.01203 |
| GO:0006820 | anion transport | 115 | 10 | 4.41 | 0.01285 |
| GO:0009267 | cellular response to starvation | 5 | 2 | 0.19 | 0.01359 |
| GO:0016036 | cellular response to phosphate starvation | 5 | 2 | 0.19 | 0.01359 |
| GO:0042594 | response to starvation | 5 | 2 | 0.19 | 0.01359 |
| GO:0050793 | regulation of developmental process | 25 | 4 | 0.96 | 0.01425 |
| GO:0051128 | regulation of cellular component organization | 26 | 4 | 1 | 0.01635 |
| GO:0005984 | disaccharide metabolic process | 46 | 5 | 1.76 | 0.03061 |
| GO:0009311 | oligosaccharide metabolic process | 47 | 5 | 1.8 | 0.03321 |
| GO:0009607 | response to biotic stimulus | 65 | 6 | 2.49 | 0.0379 |
| GO:0010208 | pollen wall assembly | 1 | 1 | 0.04 | 0.03836 |
| GO:0010584 | pollen exine formation | 1 | 1 | 0.04 | 0.03836 |
| GO:0010927 | cellular component assembly involved in morphogenesis | 1 | 1 | 0.04 | 0.03836 |
| GO:0080110 | sporopollenin biosynthetic process | 1 | 1 | 0.04 | 0.03836 |
| GO:0085029 | extracellular matrix assembly | 1 | 1 | 0.04 | 0.03836 |

## LncRNA-mRNA co-expression analysis of response to clubroot infection process

In this study, all significant differentially expressed lncRNAs and mRNAs were used to calculate the Pearson correlation coefficients based on their expression level. The top 600 potential

lncRNA-mRNA regulated pairs whose Pearson correlation coefficient greater than 0.8 were used to construct the regulatory network (S1 Fig). The 40 clubroot diseases related to mRNAs and 16 lncRNAs targeting these significantly differentially expressed mRNAs were also used to construct the correlation network of lncRNA-mRNA. In total, the resulting lncRNA:mRNA association network had 31 nodes and 19 connections between the 15 mRNAs and 16 lncRNAs (Fig 7, S5 Table). Among these molecules, most of mRNAs and lncRNAs are significantly downregulated.This regulation network indicated that four lncRNAs were predicted to be targets of 2 lncRNAs. BraA07g029760.3C and BraA07g0285503C were both targeted by lncRNA TCONS-00034121. In addition, the other three genes were all targeted by lncRNA TCONS-00049044. These results suggest that the expression profiles of mRNA and lncRNA are significantly correlated.

To elucidate the lncRNA-mRNA co-expression network, we annotated the function of the target genes by comparison with Arabidopsis. The annotation showed that they belonged to defense response proteins (66.67%), protein phosphorylation (13.33%), root hair cell differentiation (13.33%) and regulation of the salicylic acid biosynthetic process (6.67%) (Table 3). KEGG annotation showed that the vast majority of the genes involved in the biosynthesis of secondary metabolism pathways and plant-pathogen interactions.

## Discussion

In the present study, RNA–seq technology was used to investigate the global lncRNA-mRNA regulatory network between the *B. rapa* line before and after *P. brassicae* infection. The results

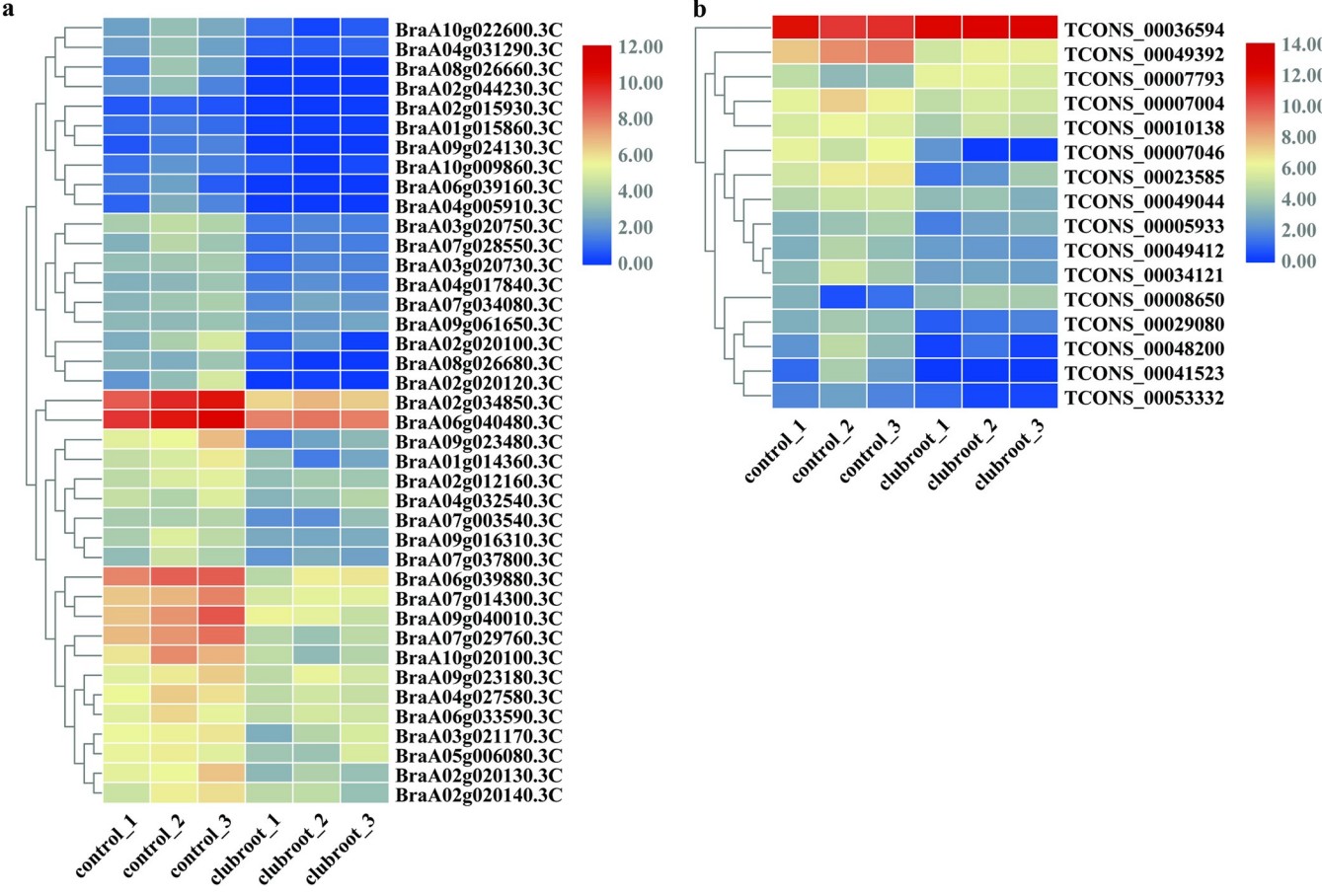

**Fig 6.** Expression levels of the 40 downregulated mRNAs (A) and 16 lncRNAs (B).

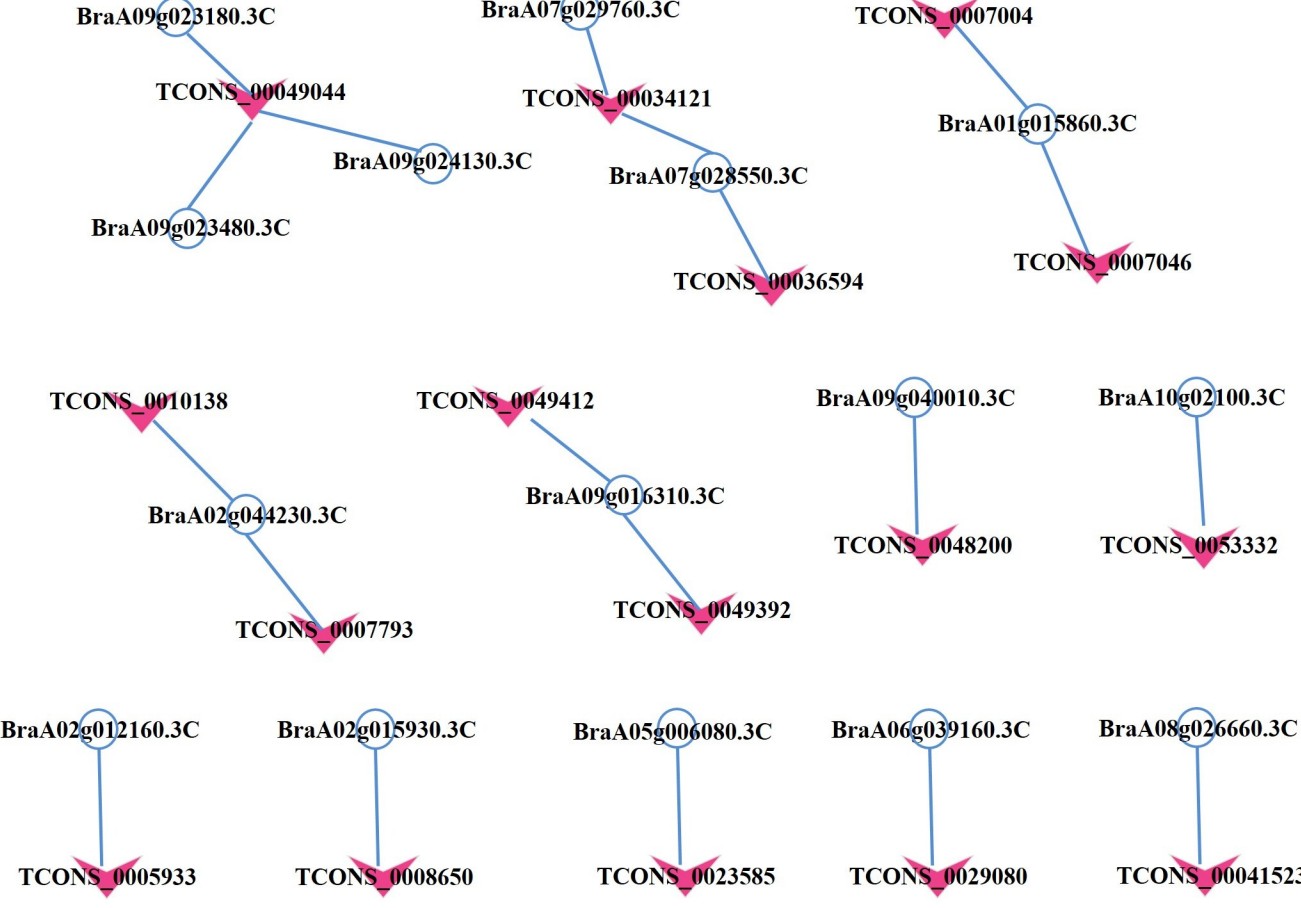

**Fig 7. LncRNA-mRNA correlation network of respose to clubroot infection process.**

of the differentially expressed analysis showed that the number of significantly differentially expressed mRNAs and lncRNAs were approximately 3 times higher in downregulated than in upregulated number. A total of 5193 and 114 mRNAs and lncRNAs were significantly differentially expressed. These results showed that a more complicated regulatory network exists in the clubroot infected plant. The GO annotation of the potential lncRNA targets showed that most upregulated significantly differentially expressed mRNAs were involved in the regulation of gene expression, and the downregulated significantly expressed mRNAs were closely related to stimulus, response to stress, defense response and response to biotic stimulus. The results of KEGG pathway analysis for the above mentioned lncRNA targets showed that they involved in the plant hormone signal transduction. The same conclusion was also reported in previous research [2, 23, 25, 61]. Jasmonic acid and salicylic acid regulate disease resistance in Arabidopsis [62]. The MAPK signaling pathway plays a pivotal role in the cellular processes such as proliferation, apoptosis, and gene regulation [63]. The metabolism of drug and xenobiotic pathway function is to oxidize small foreign organic molecules, such as toxins or drugs [64]. These results suggest that clubroot resistance and some cellular biological processes may be repressed during pathogen infection. The reaction mechanism that responds to xenobiotics may be activated during pathogen infection. Our results provide a distinct landscape in regard to the molecular mechanisms underlying *P. brassicae* infection.

**Table 3. Annotation of 15 mRNAs involved in LncRNA-mRNA co-expression network by comparison with the Arabidopsis genome.**

| Gene | Gene description | Arabidopsis | Functional annotation |
|------|------------------|-------------|----------------------|
| BraA01g015860.3C | U-box domain-containing protein 35 | AT4G25160 | protein phosphorylation, protein ubiquitination |
| BraA02g012160.3C | calmodulin-binding protein 60 B-like | AT5G57580 | regulation of salicylic acid biosynthetic process |
| BraA02g015930.3C | U-box domain-containing protein 35 | AT5G51270 | protein phosphorylation, protein ubiquitination, |
| BraA02g044230.3C | defensin-like protein 6 | AT5G63660 | defense response,<br>defense response to fungus,<br>killing of cells of other organisms, |
| BraA05g006080.3C | nematode resistance protein-like HSPRO2 | AT2G40000 | defense response to bacterium, incompatible interaction, response to oxidative stress, response to salicylic acid, tryptophan catabolic process to kynurenine |
| BraA06g039160.3C | universal stress protein PHOS32 | AT2G03720 | root hair cell differentiation, |
| BraA07g028550.3C | protein SAR DEFICIENT 1-like | AT1G73805 | cellular response to molecule of bacterial origin, defense response to bacterium, defense response to oomycetes, plant-type hypersensitive response, positive regulation of defense response to bacterium, regulation of salicylic acid biosynthetic process, regulation of systemic acquired resistance,<br>regulation of transcription, DNA-templated,<br>response to UV-B, response to bacterium |
| BraA07g029760.3C | MLP-like protein 31 | AT1G70850 | defense response |
| BraA08g026660.3C | MLP-like protein 31 | AT1G70830 | defense response |
| BraA09g016310.3C | MLO-like protein 6 | AT1G61560 | defense response, defense response to fungus,<br>incompatible interaction, response to biotic stimulus |
| BraA09g023180.3C | MLP-like protein 328 | AT2G01520 | defense response, response to phenylpropanoid, response to zinc ion, vegetative to reproductive phase transition of meristem |
| BraA09g023480.3C | defensin-like protein 1 | AT2G02130 | defense response, defense response to fungus,<br>killing of cells of other organism |
| BraA09g024130.3C | universal stress protein PHOS32-like | AT2G03720 | root hair cell differentiation |
| BraA09g040010.3C | MLP-like protein 43 | AT1G35310 | defense response |
| BraA10g020100.3C | Polyketide cyclase/ dehydrase and lipid transport superfamily protein | AT1G70860 | defense response |

Some quantitative trait loci (QTLs) related to clubroot diseases have been identified [8–10, 13, 18, 65, 66]. LncRNAs are a group of endogenous RNAs that function as regulators of gene expression, and may play an important role in several biological processes of plants [24]. LncRNA COLDAIR was reported to be required for establishing stable repressive chromatin at *FLOWERING LOCUS* [67]. LncRNA ASL can be regulated by ATRRP6L to modulate H3K27me3 levels functions in the autonomous pathway in *Arabidopsis* [68]. Therefore, we first investigated the lncRNA response to *Plasmodiophora brassicae* infection in Pakchoi and attempted to identify genes regulated by lncRNAs. The markers of QTL intervals that have been identified were mapped to the genome to examine the position relation of the QTLs and the genes (a total of 15 mRNAs and 16 lncRNAs) that were identified as related to clubroot disease in this study. The results show that lncRNA TCONS_00007793 localizes near the QTL *Anju1* region on Chromosome A02 [11], two lncRNAs (TCONS_00007004, TCONS_00007046) localize near the QTL *Rcr8*, which was identified on Chromosome A02 [18], lncRNA TCONS_00014032 localizes near the QTL *CRd*, which was identified on Chromosome A3 [65], lncRNA TCONS_00038153 localizes near the QTL CRs, which were identified on Chromosome A8 [66], lncRNAs (TCONS_00034121 and TCONS_00036594) localizes near

the QTL *qBrCR38-1*, identified by the bulked segregant analysis (BSA) method [69] on Chromosome A07, lncRNA TCONS_00041523 localized near the QTL *qBrCR38-2*, which has been identified on Chromosome A08 in the same experience. These lncRNAs associated with the QTL regions maybe have the function of regulating gene expression [70].

We investigated the expression patterns of lncRNAs and mRNAs and constructed a lncRNA-mRNA regulatory network for *P. brassicae* infected Pakchoi and control. This network can provide a global view of all possible lncRNA-coding gene expression associations based on high-through RNA-seq data. The functional annotation shows that these lncRNAs might exhibit coordinating roles towards transcriptional regulation of the defense responsive genes. KEGG annotation shows that these genes, targeted by lncRNAs, are involved in the biosynthesis of secondary metabolism pathways which are essential for many physiological processes in plants, including pathogen invasion [71]. Although many lncRNAs have been found, their biological functions remain unclear. Further research on the specific role(s) of these lncRNAs will provide additional information regarding their detailed roles in pathogen defense.

## Supporting information

**S1 Table. Primers used in this study.**
(DOC)

**S2 Table. Information of 40 significantly differentially expressed mRNAs in Fig 2A.**
(XLSX)

**S3 Table. Information of mRNAs in Fig 5.**
(XLSX)

**S4 Table. Information of the 40 mRNAs in Fig 6.**
(XLSX)

**S5 Table. Information of the co-expression network in Fig 7.**
(XLSX)

**S1 Fig. LncRNA-mRNA correlation network of the top 600 potential lncRNA-mRNA regulated pairs based on all significant differentially expressed LncRNAs and mRNAs.** The arrow and circle nodes denote lncRNA and mRNA, respectively. Each gray edge denotes a potential target relationship between a gene or lncRNA.
(PDF)

## Acknowledgments

This work was supported by Shanghai Agriculture Applied Technology Development Program, China (Grant No.G2014070108), Agriculture Research System of Shanghai, China (Grant No. 201903) and National Key R&D Program of China 2017YFD0101803. The funders had no role in study design, data collection and analysis, decision to publish, or preparation of the manuscript.

## Author Contributions

**Conceptualization:** Yuying Zhu.

**Data curation:** Hongfang Zhu, Xiaofeng Li.

**Formal analysis:** Hongfang Zhu, Wen Zhai.

**Investigation:** Hongfang Zhu, Xiaofeng Li, Dandan Xi.

**Methodology:** Hongfang Zhu, Xiaofeng Li, Zhaohui Zhang.

**Software:** Wen Zhai.

**Writing – original draft:** Hongfang Zhu, Wen Zhai.

**Writing – review & editing:** Hongfang Zhu, Yuying Zhu.

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
