## [Decision Letter · Decision Letter 0]

28 Aug 2019

PONE-D-19-15638

Intergrating long noncoding RNAs and mRNAs expression profiles of response to Plasmodiophora brassicae infection in Pakchoi (Brassica campestris ssp. chinensis Makino)

PLOS ONE

Dear Dr. Yuying Zhu,

Thank you for submitting your manuscript to PLOS ONE. After careful consideration, we feel that it has merit but does not fully meet PLOS ONE’s publication criteria as it currently stands. Therefore, we invite you to submit a revised version of the manuscript that addresses the points raised during the review process.

The topic of the manuscript is very interesting, however both the reviewers raised many criticisms that have to be addressed and substantial changes are needed to make the manuscript suitable for publication.

We would appreciate receiving your revised manuscript by Oct 12 2019 11:59PM. To enhance the reproducibility of your results, we recommend that if applicable you deposit your laboratory protocols in protocols.io, where a protocol can be assigned its own identifier (DOI) such that it can be cited independently in the future. For instructions see: http://journals.plos.org/plosone/s/submission-guidelines#loc-laboratory-protocols

We look forward to receiving your revised manuscript.

Kind regards,

Serena Aceto, Ph.D.

Academic Editor

PLOS ONE

Journal Requirements:

2. We note that you are reporting an analysis of a microarray, next-generation sequencing, or deep sequencing data set. PLOS requires that authors comply with field-specific standards for preparation, recording, and deposition of data in repositories appropriate to their field. Please upload these data to a stable, public repository (such as ArrayExpress, Gene Expression Omnibus (GEO), DNA Data Bank of Japan (DDBJ), NCBI GenBank, NCBI Sequence Read Archive, or EMBL Nucleotide Sequence Database (ENA)). In your revised cover letter, please provide the relevant accession numbers that may be used to access these data. For a full list of recommended repositories, see http://journals.plos.org/plosone/s/data-availability#loc-omics or http://journals.plos.org/plosone/s/data-availability#loc-sequencing.

3. Please amend the manuscript submission data (via Edit Submission) to include author Dandan Xi.

4. Please amend your authorship list in your manuscript file to include author Wen Zhai.

Reviewers' comments:

Reviewer's Responses to Questions

**Comments to the Author**

1. Is the manuscript technically sound, and do the data support the conclusions?

Reviewer #1: Yes

Reviewer #2: No

2. Has the statistical analysis been performed appropriately and rigorously? 

Reviewer #1: I Don't Know

Reviewer #2: N/A

3. Have the authors made all data underlying the findings in their manuscript fully available?

Reviewer #1: Yes

Reviewer #2: Yes

4. Is the manuscript presented in an intelligible fashion and written in standard English?

Reviewer #1: No

Reviewer #2: No

5. Review Comments to the Author

Reviewer #1: This manuscript describes the changes in the expression of protein coding genes (mRNA) as well as those encoding long non-coding (lnc)RNAs in Brassica rapa in response to infection with the clubroot causing pathogen, Plasmodiophora brassicae. It attempts to integrate the findings with respect to lncRNA expression and mRNA expression.

Experiments have been designed and conducted appropriately although there are some significant issues. Please see major comments below.

The manuscript has been generally written well but there are still many issues--see minor comment for a non-exhaustive list of errors. It is impossible for me to list all the errors and the authors should ensure that the submission is free from these types of mistakes.

Major comments:

1. The samples were collected 6 weeks after sowing when full-blown root galls had developed. I do not understand the logic of investigating molecular responses after the development of galls. It would have been more revealing at earlier stages of pathogenesis. In my opinion, earlier stages in the disease should be sampled and mRNA and lncRNA profiles investigated.

2.Were there any pathogen mRNA and lncRNA detected? This is such a late stage that this may have been possible. IF so, it should be reported and if not, a statement should be made to that effect.

3. Line 376, are you talking about QTLs associated with resistance? However, in this case you used susceptible lines. This part of the discussion is confusing and should be clarified.

4. The Discussion section is extremely weak, or non-existent. It has be significantly strengthened to discuss their findings and integrating them with the information that is available in the literature. Currently this section is a sum total of 31 lines or just about one page. The most important part of these types of descriptive articles is their discussion about possible biological relevance of their findings. This should be rewritten in its entirety.

5. Why does the title of the manuscript say Brassica campestris? I thought that Pakchoi is Brassica rapa? Please be accurate and consistent.

Minor comments:

1. Some typographical and grammatical errors in the abstract include, line 11 “researches”; line 12 “works”; line 13 “….profiles of response to…”.

2. Line 28, “…a kind of soil-borne disease…”. It is a soil-borne disease, NOT “a kind-of soil-borne disease”.

3. Line 36, not “clubroot resistance crops” but “clubroot resistant crops”.

4. Line 32, Brassica rapa????

5. Lines 53 and 60, specify the Brassica species! In general, please specify the Brassica (or any plant) species when you are talking about the results from another study.

6. Line 69—do not begin a sentence with “And”.

7. Line 79, wrong reference format.

8. Line 122, NOT “reversely” transcribed.

9. Line 181, should be “Functional” enrichment.

10. Line 281, should be “Functional” annotation.

11. Line 345, “Pearson” correlation coefficient NOT “person”.

Reviewer #2: I have carefully read the manuscript PONE-D-19-15638 “Integrating long noncoding RNAs and mRNAs expression profiles of response to Plasmodiophora brassicae infection in Pakchoi (Brassica campestris ssp. Chinensis Makino)” by Zhu et al. The topic of the manuscript is interesting and the integration of lncRNAs to increase our understanding of clubroot disease is timely and novel. However, there are some shortcomings of the manuscript in its present form most notably the absence of an acceptable discussion of the results. More detailed comments can be found below.

The English of the manuscript is understandable, but I would strongly advice the authors to consult someone proficient in scientific English for language editing as there are several grammatical errors in the MS. There are many sloppy errors (inconsistent spelling, spelling mistakes, etc) throughout the text. I have pointed out some issues in the detailed comments below, but this list is not exhaustive.

Overall the manuscript would add important findings to our understanding of clubroot disease, but I strongly recommend that the authors considerably revise the manuscript to highlight and describe the very interesting findings they collected.

Detailed comments:

Abstract:

L 11: delete “Although lots of researches have been conducted during past decades” – grammer errors and IMHO this is not important in the abstract.

L17: specify which type of enrichment analyses

L19: define which type of interaction relationship, list the most important groups of interactions to provide more info for the reader on the biological relevance of those rather than just providing numbers.

L21: change “15 clubroot disease related” into “15 P. brassicae mRNAs”, maybe provide more detail on these RNAs.

Introduction

The introduction informative about the aims and guides the reader to the topic of the manuscript. Some phrases are difficult to understand, especially if the reader is not familiar with the original literature cited, therefore I would advise the authors to carefully edit the introduction.

L29: no need to write P. brassicae in brackets.

L60: “between CR and clubroot-resistant (CS) lines” this sentence is not clear, should this be clubroot susceptible (CS)?

L60: The sentence starting with “It was also found….” is unclear, please rephrase and explain what you mean with “updated SA function”.

Material and Methods

L102ff: Sample collection: the authors state that they use race 7 of P. brassice (also check for consistent spelling!). Was this a single spore isolate or a field population showing the characteristics of race 7 using the Williams differential? For all further experiments it will make a huge difference if the experiments were conducted using a single spore isolate or a field population.

L105ff: when were the plants inoculated and with approximately which amount of P. brassicae spores?

L107: please change “6 weeks after sowing” by the days post inoculation as this will provide more information on the disease progression

L129ff: what does “clean data” and “dirty reads” refer to? Please rephrase this and avoid laboratory jargon. Having a flow chart of the procedure including versions of the software in the Supplement is usually very helpful for such methods sections (also for LncRNA).

L172: state the type of mastermix used

L181ff: This section is not fully clear, please provide more information on the statistical analyses (R packages, settings, assumptions etc used in the analyses)

L188: The sentence “the p value…” is incomplete, please rephrase

L193ff: this section is not clear.

L193: lncRNAs - please make sure the correct spelling throughout the MS

Results

L217: use control or C instead of CK, this abbreviation is not intuitive.

L223: use “the number of reads after qality filtering (or quality control) were….” Instead of “clean reads”

L225: the sentence startin with Q30 is incomplete

L293: “these findings indicate that mRNAs were participating in the defence of clubroot” – This sentence is not correct. There is not evidence that the mRNAs themselves are involved in clubroot defence, rather it can be assumed that the genes/proteins the mRNA codes for are involved in the reaction. Please rephrase.

L301: The phrase “previous researches” is odd, rephrase to “previous studies” or similar.

L307: why were the GO terms restricted to “biological process” and why were molecular function and cellular component omitted? Is there a reason for this choice?

L346: is it really “person correlation coefficient” or should it be Pearson’s

Tables and Figures:

Overall the figure and table capitations could be a bit more informative and descriptive. Not every image can be easily understood, therefore some comments on specific issues:

Tables: Please describe what the different values stand for. I suspect that “annotated” means the number of transcripts that were assigned this GO term in the full dataset – or does this refer to the number of transcripts that were upregulated in clubroot tissue and assigned a certain GO term? Significant – is the number of these GO terms that were significant in which respect – significantly up/downregulated? What does “expected” mean and where does this come from? Classic Fisher refers to what – the significance of the GO-term, the up/downregulated GO-term, the transcripts?

Fig 2, 5, 6 Heatmap figures: can you provide any biological information to the genes other than the BRA accession? Maybe adding some sort of functional annotation (GO term, gene name, function of the gene,….). Which values are displayed? FPKM values, and if yes were these normalised? DEGs – but then which values were compared to give the values?

Fig3A: what do the numbers on the x-axis refer to? Please describe in the figure capitation or in the image.

Fig7: This figure is not very informative in the present form and also its not a network but a series of correlated genes. Please add information and annotations (which genes are we looking at?) to the figure or convert it into a table (which would provide more information on the individual correlations)

Fig S1: please provide a description of the figure. It is nearly impossible to understand this figure the way it is currently presented.

Discussion:

Unfortunately, the discussion feels very incomplete especially as the authors present a number of fascinating results. The authors fail to discuss what the findings of the correlation of lncRNAs and mRNAs presented mean for the biology of clubroot disease. The concept of lncRNAs is employed to clubroot for the first time, so there are plenty of factors that can be discussed and described here. Also there is no comparison to other transcriptomics studies of which there are plenty on a multitude of clubroot hots, resistant and susceptible interactions, on the intraplant variation etc. Please use this pool of references to discuss the results in a broader context.

L357ff: the first paragraph of the discussion is mostly results. Please move the description of the KEGG analyses into the results section, where only GO terms are described currently. Many of these processes have been identified in previous transcriptomic studies of clubroot, please cite those studies and compare their results to the ones generated in this study.

L376ff: This information is interesting, but most of the data are not yet available. Therefore the validity of more than half of the second paragraph of the discussion cannot be assessed.

Data availability

Data are available.

6. PLOS authors have the option to publish the peer review history of their article (what does this mean?). If published, this will include your full peer review and any attached files.

Reviewer #1: No

Reviewer #2: No

---

## [Author Response · Author response to Decision Letter 0]

16 Oct 2019

Reviewer #1

1. This manuscript describes the changes in the expression of protein coding genes (mRNA) as well as those encoding long non-coding (lnc)RNAs in Brassica rapa in response to infection with the clubroot causing pathogen, Plasmodiophorabrassicae. It attempts to integrate the findings with respect to lncRNA expression and mRNA expression.Experiments have been designed and conducted appropriately although there are some significant issues. Please see major comments below.The manuscript has been generally written well but there are still many issues--see minor comment for a non-exhaustive list of errors. It is impossible for me to list all the errors and the authors should ensure that the submission is free from these types of mistakes.

Response: We appreciated reviewer’s interest in our work. We have corrected these issues in the revised manuscript according to the reviewers’ comments.

2. The samples were collected 6 weeks after sowing when full-blown root galls had developed. I do not understand the logic of investigating molecular responses after the development of galls. It would have been more revealing at earlier stages of pathogenesis. In my opinion, earlier stages in the disease should be sampled and mRNA and lncRNA profiles investigated.

Response: All plants were sown in a pot containing 5×106 spores per gram of dry soil.In 6 weeks after sowing, the plant has grown to 4or5 leaves. In this development stage, infected root showed enlargement. Therefore, our study sampled in the late stage of disease. Please refer to line111-115.

3.Were there any pathogen mRNA and lncRNA detected? This is such a late stage that this may have been possible. IF so, it should be reported and if not, a statement should be made to that effect.

Response: Yes, we actually find pathogen mRNA in our samples. About 3.68% reads of CS22A(infected root) blast to Plasmodiophorabrassicae genome.

4. Line 376, are you talking about QTLs associated with resistance? However, in this case you used susceptible lines. This part of the discussion is confusing and should be clarified.

Response: This is a very good point. Some quantitative trait loci (QTLs) which are related to clubroot diseases have been identified. The lncRNAs are a group of endogenous RNAs that function as regulators of gene expression and may play an important role in several biological processes of plants. So we want to find some negative or positive correlation of lncRNAs and mRNAs. Most of the 15 mRNAs are belonged to defense response proteins by compared with Arabidopsis. However, little is known about the 16 lncRNAs. We agree that further research on these lncRNAs will provide additional information about their detailed roles in pathogen defense.

5. The Discussion section is extremely weak, or non-existent. It has be significantly strengthened to discuss their findings and integrating them with the information that is available in the literature. Currently this section is a sum total of 31 lines or just about one page. The most important part of these types of descriptive articles is their discussion about possible biological relevance of their findings. This should be rewritten in its entirety.

Response: The Discussion sectionhas been thoroughly rewritten and the confusing sentence has been corrected. Please refer to line 418-476.

6. Why does the title of the manuscript say Brassica campestris? I thought that Pakchoi is Brassica rapa? Please be accurate and consistent.

Response: Pakchoi is also named non-heading Chinese cabbage. There have been a lot of studies in the past that have named Brassica campestris, such as Du et al.2008, Ma et al. 2010, Zhu et al. 2017, Fan et al 2019.

Du S and Y. Zhang, et al. (2008)."Regulation of nitrate reductase by nitric oxide in Chinese cabbage pakchoi (Brassica chinensis L.)." Plant Cell Environ 31 (2): 195-204.

Ma, J. and X. Hou, et al. "Cloning and Characterization of the BcTuR3 Gene Related to Resistance to Turnip Mosaic Virus (TuMV) from Non-heading Chinese Cabbage." Plant Molecular Biology Reporter. 2010, 28 (4): 588-596.

Fan, X., Xue, F., Song, B., et al. Effects of Blue and Red Light On Growth And Nitrate Metabolism In Pakchoi. Open Chemistry, 2019, 17(1):456-464.

Zhu H, Li X, Zhai W, et al. Effects of low light on photosynthetic properties, antioxidant enzyme activity, and anthocyanin accumulation in purple pak-choi(Brassica campestris ssp. Chinensis Makino).Plos One, 2017, 12(6):e0179305.

7. Some typographical and grammatical errors in the abstract include, line 11 “researches”; line 12 “works”; line 13 “….profiles of response to…”.

Response: We have corrected this problem in the revised manuscript. Please refer to line 11-12.

8. Line 28, “…a kind of soil-borne disease…”. It is a soil-borne disease, NOT “a kind-of soil-borne disease”.

Response: This sentence has been corrected. Please refer to line 32. 

9. Line 36, not “clubroot resistance crops” but “clubroot resistant crops”.

Response:This sentence has been corrected. Please refer to line 41.

10. Line 32, Brassica rapa????

Response: Pakchoi is categorized as Brassica campestris ssp. Chinensis Makino, whose genome is similar to Chinese cabbage (Brassica rapa. L).

11. Lines 53 and 60, specify the Brassica species! In general, please specify the Brassica (or any plant) species when you are talking about the results from another study.

Response:This issue has been corrected. We have specified the brassica species. Please refer to line 56-77.

12. Line 69—do not begin a sentence with “And”.

Response: We have corrected this problem. Please refer to line 58.

13. Line 79, wrong reference format.

Response:We have corrected reference format. Please refer to line 70.

14. Line 122, NOT “reversely” transcribed.

Response: This issue has been corrected. Please refer to line 128.

15. Line 181, should be “Functional” enrichment.

Response: This issue has been corrected. Please refer to line 191.

16. Line 281, should be “Functional” annotation.

Response: This issue has been corrected. Please refer to line 298.

17. Line 345, “Pearson” correlation coefficient NOT “person”.

Response: We have corrected it in the revised manuscript. Please refer to line 389.

Reviewer #2: 

1. I have carefully read the manuscript PONE-D-19-15638 “Integrating long noncoding RNAs and mRNAs expression profiles of response to Plasmodiophorabrassicae infection in Pakchoi (Brassica campestris ssp. Chinensis Makino)” by Zhu et al. The topic of the manuscript is interesting and the integration of lncRNAs to increase our understanding of clubroot disease is timely and novel. However, there are some shortcomings of the manuscript in its present form most notably the absence of an acceptable discussion of the results. More detailed comments can be found below.The English of the manuscript is understandable, but I would strongly advice the authors to consult someone proficient in scientific English for language editing as there are several grammatical errors in the MS. There are many sloppy errors (inconsistent spelling, spelling mistakes, etc) throughout the text. I have pointed out some issues in the detailed comments below, but this list is not exhaustive.

Overall the manuscript would add important findings to our understanding of clubroot disease, but I strongly recommend that the authors considerably revise the manuscript to highlight and describe the very interesting findings they collected.

Response:We appreciated reviewer’s interest in our work. The manuscript has been thoroughly rewritten according to the reviewers’ comments.

2.L 11: delete “Although lots of researches have been conducted during past decades” – grammer errors and IMHO this is not important in the abstract.

Response: This sentence has been deleted.

3.L17: specify which type of enrichment analyses

Response: We have corrected it in the revised manuscript. Please refer to line 16.

4.L19: define which type of interaction relationship, list the most important groups of interactions to provide more info for the reader on the biological relevance of those rather than just providing numbers.

Response: This issue has been corrected and we have also provide more detail on the 15 RNAs in the Results section as the reviewer suggested. Please refer to line 385-401 and S5 table.

5.L21: change “15 clubroot disease related” into “15 P. brassicae mRNAs”, maybe provide more detail on these RNAs.

Response: This issue has been corrected and we have also provide more detail on these RNAs in the Results section as the reviewer suggested. Please refer to line 403-409.

6.Introduction.The introduction informative about the aims and guides the reader to the topic of the manuscript. Some phrases are difficult to understand, especially if the reader is not familiar with the original literature cited, therefore I would advise the authors to carefully edit the introduction.

Response: We agree and have revised the section of Introduction accordingly. Please refer to line 32-100.

7.L29: no need to write P. brassicae in brackets.

Response: We have corrected this problem in the revised version of the manuscript. Please refer to line 34.

8.L60: “between CR and clubroot-resistant (CS) lines” this sentence is not clear, should this be clubroot susceptible (CS)?

Response:We have corrected this problem in the revised version of the manuscript. Please refer to line 65.

9.L60: The sentence starting with “It was also found….” is unclear, please rephrase and explain what you mean with “updated SA function”.

Response: We have corrected this problem in the revised version of the manuscript. Please refer to line 66.

10.L102ff: Sample collection: the authors state that they use race 7 of P. brassice (also check for consistent spelling!). Was this a single spore isolate or a field population showing the characteristics of race 7 using the Williams differential? For all further experiments it will make a huge difference if the experiments were conducted using a single spore isolate or a field population.

Response:This was corrected in the Materials and Methods section. Until 2017, 39 counties and 9 towns of Shanghai had a breakout of clubroot disease and the affected area had reached 2500 hm2. The race 7 of P. brassice was characrilized by field population come from disease nurseries in Qingpu district of Shanghai. Some other scientific institutions have also identified the same result, such as East China University of Science and Technology and Chinese Academy of Agricultural Sciences.

11.L105ff: when were the plants inoculated and with approximately which amount of P. brassicae spores?

Response:We have added this information in the Materials and Methods section of the revised manuscript. Please refer to line 111-115.

12.L107: please change “6 weeks after sowing” by the days post inoculation as this will provide more information on the disease progression

Response:We have corrected it in the revised manuscript. Please refer to line 115.

13.L129ff: what does “clean data” and “dirty reads” refer to? Please rephrase this and avoid laboratory jargon. Having a flow chart of the procedure including versions of the software in the Supplement is usually very helpful for such methods sections (also for LncRNA).

Response:We have included this information in the Materials and Methods section of the revised manuscript. Please refer to line 135-140.

14.L172: state the type of mastermix used

Response:We have corrected this problem. Please refer to line 181.

15.L181ff: This section is not fully clear, please provide more information on the statistical analyses (R packages, settings, assumptions etc used in the analyses)

Response:We have included this information in the Materials and Methods section of the revised manuscript. Please refer to line 134-173.

16.L188: The sentence “the p value…” is incomplete, please rephrase

Response: This sentence has been corrected. Please refer to line 198.

17.L193ff: this section is not clear.

Response:Please refer to line204.

18.L193: lncRNAs - please make sure the correct spelling throughout the MS

Response:This sentence has been corrected. Please refer to line 204.

19.L217: use control or C instead of CK, this abbreviation is not intuitive.

Response: We have corrected this problem in the revised version of the manuscript. Please refer to line 231.

20.L223: use “the number of reads after qality filtering (or quality control) were….” Instead of “clean reads”

Response:This sentence has been removed from the revised manuscript. Please refer to line 236.

21.L225: the sentence startin with Q30 is incomplete

Response:We have corrected this problem in the revised version of the manuscript. Please refer to line 239.

22.L293: “these findings indicate that mRNAs were participating in the defence of clubroot” – This sentence is not correct. There is not evidence that the mRNAs themselves are involved in clubrootdefence, rather it can be assumed that the genes/proteins the mRNA codes for are involved in the reaction. Please rephrase.

Response: We have corrected this problem in the revised version. Please refer to line 333.

23.L301: The phrase “previous researches” is odd, rephrase to “previous studies” or similar.

Response:This issue has been corrected. Please refer to line 341.

24.L307: why were the GO terms restricted to “biological process” and why were molecular function and cellular component omitted? Is there a reason for this choice?

Response: We have corrected this problem in the revised version of the manuscript. Please refer to line 347-370.

25.L346: is it really “person correlation coefficient” or should it be Pearson’s

Response: This issue has been corrected. Please refer to line 389.

26. Overall the figure and table capitations could be a bit more informative and descriptive. Not every image can be easily understood, therefore some comments on specific issues: Tables: Please describe what the different values stand for. I suspect that “annotated” means the number of transcripts that were assigned this GO term in the full dataset – or does this refer to the number of transcripts that were upregulated in clubroot tissue and assigned a certain GO term? Significant – is the number of these GO terms that were significant in which respect – significantly up/downregulated? What does “expected” mean and where does this come from? Classic Fisher refers to what – the significance of the GO-term, the up/downregulated GO-term, the transcripts?

Response:We have added more information and description on the figures and tables according to the reviewers’ comments.

27.Fig 2, 5, 6 Heatmap figures: can you provide any biological information to the genes other than the BRA accession? Maybe adding some sort of functional annotation (GO term, gene name, function of the gene,….). Which values are displayed? FPKM values, and if yes were these normalised? DEGs – but then which values were compared to give the values?

Response:We have corrected Figures show the nomalised significant differentially expressed mRNA and lncRNA.

28.Fig3A: what do the numbers on the x-axis refer to? Please describe in the figure capitation or in the image.

Answer: we have added the description on the x-axis in the figure capitation.

29.Fig7: This figure is not very informative in the present form and also its not a network but a series of correlated genes. Please add information and annotations (which genes are we looking at?) to the figure or convert it into a table (which would provide more information on the individual correlations).

Response:We have corrected Figure 7 and added the Supplemental Table3 containing detailed data used to create this figure.

30.Fig S1: please provide a description of the figure. It is nearly impossible to understand this figure the way it is currently presented.

Response: We have added the description of the figure in the figure capitation.

31.Unfortunately, the discussion feels very incomplete especially as the authors present a number of fascinating results. The authors fail to discuss what the findings of the correlation of lncRNAs and mRNAs presented mean for the biology of clubroot disease. The concept of lncRNAs is employed to clubroot for the first time, so there are plenty of factors that can be discussed and described here. Also there is no comparison to other transcriptomics studies of which there are plenty on a multitude of clubroothots, resistant and susceptible interactions, on the intraplant variation etc. Please use this pool of references to discuss the results in a broader context.

Response:We agree that this section, as it was written, was confusing and problematic. We have thoroughly revised this part of the Dicussion and compared our results with other transcriptomics studies to increase the understandiong of clubroot disease. Although lots of lncRNAs have been found, their biological functions remain unclear. We believe that several more studies will be needed to elucidate these issues. .

32.L357ff: the first paragraph of the discussion is mostly results. Please move the description of the KEGG analyses into the results section, where only GO terms are described currently. Many of these processes have been identified in previous transcriptomic studies of clubroot, please cite those studies and compare their results to the ones generated in this study.

Response:We have corrected these issues in the revised manuscript according to the reviewers’ comments.

33.L376ff: This information is interesting, but most of the data are not yet available. Therefore the validity of more than half of the second paragraph of the discussion cannot be assessed.

Response: Some quantitative trait loci (QTLs) which are related to clubroot diseases have been identified. The lncRNAs are a group of endogenous RNAs that function as regulators of gene expression and may play an important role in several biological processes of plants. So we want to find some negative or positive correlation of lncRNAs and mRNAs. Most of the 15 mRNAs are belonged to defense response proteins by compared with Arabidopsis. However, little is known about the 16 lncRNAs. We agree that further research on these lncRNAs will provide additional information about their detailed roles in pathogen defense.

---

## [Decision Letter · Decision Letter 1]

25 Oct 2019

Integrating long noncoding RNAs and mRNAs expression profiles of response to Plasmodiophorabrassicaeinfection in Pakchoi (Brassica campestris ssp. chinensisMakino)

PONE-D-19-15638R1

Dear Dr. Yuying Zhu,

We are pleased to inform you that your manuscript has been judged scientifically suitable for publication and will be formally accepted for publication once it complies with all outstanding technical requirements.

With kind regards,

Serena Aceto, Ph.D.

Academic Editor

PLOS ONE

Additional Editor Comments (optional):

Reviewers' comments:

Reviewer's Responses to Questions

**Comments to the Author**

1. If the authors have adequately addressed your comments raised in a previous round of review and you feel that this manuscript is now acceptable for publication, you may indicate that here to bypass the “Comments to the Author” section, enter your conflict of interest statement in the “Confidential to Editor” section, and submit your "Accept" recommendation.

Reviewer #1: All comments have been addressed

2. Is the manuscript technically sound, and do the data support the conclusions?

Reviewer #1: Yes

3. Has the statistical analysis been performed appropriately and rigorously? 

Reviewer #1: I Don't Know

4. Have the authors made all data underlying the findings in their manuscript fully available?

Reviewer #1: Yes

5. Is the manuscript presented in an intelligible fashion and written in standard English?

Reviewer #1: Yes

6. Review Comments to the Author

Reviewer #1: The authors have addressed my comments.

7. PLOS authors have the option to publish the peer review history of their article (what does this mean?). If published, this will include your full peer review and any attached files.

Reviewer #1: No

---

## [Editor Report · Acceptance letter]

21 Nov 2019

PONE-D-19-15638R1 

Integrating long noncoding RNAs and mRNAs expression profiles of response to *Plasmodiophora brassicae* infection in Pakchoi (*Brassica campestris* ssp. *chinensis* Makino) 

Dear Dr. Zhu:

I am pleased to inform you that your manuscript has been deemed suitable for publication in PLOS ONE. Congratulations! Your manuscript is now with our production department. 

With kind regards,

on behalf of

Dr Serena Aceto 

Academic Editor

PLOS ONE